# Education Policy for Migrant Children in Thailand and How It Really Happens; A Case Study of Ranong Province, Thailand

**DOI:** 10.3390/ijerph16030430

**Published:** 2019-02-01

**Authors:** Titiporn Tuangratananon, Rapeepong Suphanchaimat, Sataporn Julchoo, Pigunkaew Sinam, Weerasak Putthasri

**Affiliations:** 1International Health Policy Program, Ministry of Public Health, Tiwanon Road, Nonthaburi 11000, Thailand; Rapeepong@ihpp.thaigov.net (R.S.); Sataporn@ihpp.thaigov.net (S.J.); Pigunkaew@ihpp.thaigov.net (P.S.); 2Department of Health, Ministry of Public Health, Tiwanon Road, Nonthaburi 11000, Thailand; 3Bureau of Epidemiology, Department of Disease Control, Ministry of Public Health, Tiwanon Road, Nonthaburi 11000, Thailand; 4National Health Commission office, National Health Building, Nonthaburi 11000, Thailand; Weerasak@nationalhealth.or.th

**Keywords:** migrants, Migrant Learning Center, education, school health, intersectionality

## Abstract

Health and education are interrelated, and it is for this reason that we studied the education of migrant children. The Thai Government has ratified ‘rights’ to education for all children in Thailand since 2005. However, there are gaps in knowledge concerning the implementation of education policy for migrants, such as whether and to what extent migrant children receive education services according to policy intentions. The objective of this study is to explore the implementation of education policy for migrants and the factors that determine education choices among them. A cross-sectional qualitative design was applied. The main data collection technique was in-depth interviews with 34 key informants. Thematic analysis with an intersectionality approach was used. Ranong province was selected as the main study site. Results found that Migrant Learning Centers (MLCs) were the preferable choice for most migrant children instead of Thai Public Schools (TPSs), even though MLCs were not recognized as formal education sites. The main reason for choosing MLCs was because MLCs provided a more culturally sensitive service. Teaching in MLCs was done in Myanmar’s language and the MLCs offer a better chance to pursue higher education in Myanmar if migrants migrate back to their homeland. However, MLCs still face budget and human resources inadequacies. School health promotion was underserviced in MLCs compared to TPSs. Dental service was underserviced in most MLCs and TPSs. Implicit discrimination against migrant children was noted. The Thai Government should view MLCs as allies in expanding education coverage to all children in the Thai territory. A participatory public policy process that engages all stakeholders, including education officials, health care providers, Non-Governmental Organizations (NGOs), MLCs’ representatives, and migrants themselves is needed to improve the education standards of MLCs, keeping their culturally-sensitive strengths.

## 1. Introduction

In 2015, 244 million people worldwide lived outside their country of birth; 31 million of them were children and 12 million were living in Asia. Since 1990, although the proportion of international migrant children has remained stable at around 1% of the world’s population, an increasing global population means an increase in the absolute number [1]. Matters surrounding migrant children are of political concern as they are a vulnerable population and prone to abuse, exploitation and human trafficking [2]. This vulnerability is more pronounced in many countries that limit ‘fundamental rights’, such as access to education and health services, to migrant children, particularly those with precarious citizenship or immigration status [3].

Surrounded by prolonged political conflict and economic uncertainty in neighbouring countries, Thailand is one of the most important destinations for migrants in the Southeast Asian region. So far, there are approximately over 3 million migrant workers from Cambodia, Lao PDR and Myanmar (so-called CLM migrants) in Thailand. The majority of these people crossed the Thai border without valid travel documents as so-called undocumented migrants [4]. Migrant workers along with their dependents including spouses, children and other close relatives. Recent estimates by Kantayaporn et al. (2013) showed that there are around 250,000–290,000 migrant children residing in Thailand [5].

The Thai Government has long advocated for fundamental rights for children, especially in terms of education and health. Education and health are strongly interrelated. Higher education levels are likely bringing knowledge to an individual so that they can better protect and promote their health. Individuals with a better health status have a greater chance of enjoying more advanced education [6]. However, these two components become more complex when dealing with migrant children and even more complex when considering that these children have unsecured citizenship status.

One of the most distinct measures to promote the wellbeing of migrant children is the health insurance policy for dependants of migrant workers, endorsed in 2013. Migrant children in Thailand are eligible for the public insurance schemes through purchasing a health insurance card issued by the Ministry of Public Health (MOPH). The card costs around USD 11 for a child under 7 years of age [7].

Some studies documented the successes and challenges of the health insurance card policy for migrants in Thailand; for example, a study by Harris (2013) and Suphanchaimat (2017) [8,9]. However, literature exploring the challenges of education services for migrants is quite sparse and this paper hopes to address, to some extent, this critical gap in knowledge.

A brief history of migrant education policies can be traced back to the early 1990s. Thailand agreed and signed the Convention on the Rights of the Child (CRC) in 1992. The country is obliged to provide non-discriminatory protection of the health and social welfare of children regardless of their ethnicity and nationality. In reality, it seems that migrant children were not on the political radar until 2005. This was the first time that policy to promote and protect social rights for migrant children materialized. According to the Cabinet Resolution on 5 July, 2005, all ‘non-Thai’ children are eligible to enjoy basic education (grade 1–9) in public Thai schools. Schools are reimbursed the education fee for each migrant child from the Thai Government based on a specified rate and equal to a Thai child [10]. However, the above resolution alone is insufficient to fulfil the needs of migrant families, and other schools or learning centres, or ‘Migrant Learning Centres’ (MLCs,) are run independently by charitable institutes or non-government organizations (NGOs) in many provinces. Among many migrant families, it seems that MLCs are more attractive than Thai public schools (TPSs) [11,12]. This point is explored in further detail in the later sections of this paper.

How migrant children in Thailand interact with the education services in Thailand has not been thoroughly explored. Gaps exist in knowledge concerning the implementation of education policy for migrants and whether and to what extent they receive services according to the policy intentions. The objective of this study is to explore the implementation of the education policy for migrants and factors that determine education choices among them. Although the main focus of this paper is about education, it also considers education related to health activities, such as health promotion in schools. An intersectionality approach, taking into account interactions between various factors such as the intrinsic characteristics of individuals (sex, ethnic, financial status, etc.) and the external environment (social structures, social values and perceptions of surrounding individuals towards migrants, etc.), is employed [13]. It is hoped that a better understanding in this area will help improve the implementation of education policies for migrants and ultimately promote the wellbeing of migrant children in Thailand in ways that suit the Thai local context.

## 2. Materials and Methods 

### 2.1. Study Design and Study Site 

This study employed a qualitative cross-sectional design by using Ranong province as a case study. The province is located in the southern region of Thailand with a geographical coverage of 3298 km^2^. It has a long natural border (approximately 95 km) on the west, connected with Myanmar [14]. Ranong has the highest proportion of migrant workers to native Thai residents (around 20%) compared to other provinces [15]. According to the Ranong Provincial Employment Office Report (2017), there are 11,441 migrant workers in the city centre, Mueang district. It is the most migrant-populated district in Ranong province, followed by Kraburi district. Accordingly, Mueang district and Kraburi district within Ranong province were purposively selected as study sites. 

### 2.2. Study Participants

As this study focuses on health and education services, especially in MLCs, most of the respondents were local providers and teachers working in the health and education fields. For the education field, to date, there are 13 MLCs in Ranong province; all of them are situated in Mueang district. The research team discussed with local providers and asked them to guide the team to the MLCs willing to participate in the study and where there were a large number of migrant students. To this end, two MLCs were selected; this number was chosen according to the feasibility of time and human resources during the course study. Some migrant children also attend TPSs and two TPSs in Mueang district and a further two in Kraburi district were selected. These TPSs were chosen based on snowball sampling after discussing with local providers. These four TPSs are located in migrant-populated communities and contain a higher number of migrant students compared to other TPSs. NGOs, MLC teachers and TPSs directors were selected to be key interviewees. The research team also conducted interviews with executive staff of the Provincial Education Office in order to gain insights into the education policy for migrants in the province as a whole.

For the health aspect, local providers working in sub-district health centres were recruited to be key informants. Executive staff members from the Provincial Public Health Office were also interviewed to shed light on the health policy towards migrants in the entire province. The team also performed some interviews with two migrant families living near the MLCs in order to gain users’ perspectives on health and education policies in Thailand. Local NGOs helped the research team approach those migrant families, and NGOs themselves also served as key informants for this study.

In summary, 34 interviewees participated in this study, with details being shown in Table 1.

### 2.3. Data Collection and Analysis

In-depth interviews were used as the main data collection techniques. The research team also conducted document reviews on relevant issues, such as financial information and school pupil profiles from MLCs, TPSs and sub-district health centres. This also served as a triangulation on the validity of the interview data. Each interview lasted about 45 min. All interviewees were conducted at the interviewees’ workplace, except for migrant parents where the interviews were performed at their households. The interviews were audiotaped and transcribed verbatim. The lead author served as the main interviewer and one to two research assistants served as note takers. Question guides for the interviews are presented in Table 2. However, in real practice, these questions were adapted to suit the interviewees’ roles in migrant education and health policies. All Myanmar participants were interviewed in the Myanmar language. Translators assisted when appropriate. 

Thematic analysis with inductive coding was performed and is presented in Table 3. The emerging codes are presented in the ‘Results’ section. The cross-cutting constructs/themes are presented in the ‘Discussion’ section, and are discussed against the concept and theory of ‘intersectionality’ [13]. 

### 2.4. Ethics Consideration

This study followed standard research protocol. The ethics approval was obtained from the Institute of Human Research Development (letter head No. IHRP 893/2560). Signed consent was obtained from all interviewees. The only exception was migrant parents where only verbal consent sufficed. The reason for obtaining verbal consent only was that the research team tried to avoid a sense of coercion over migrants. The interviewees were free to withdraw themselves at any time if there were any uncomfortable feelings. All interviewee names were kept anonymous to protect their confidentiality. 

## 3. Results

Four emerging themes emerged from the fieldwork: (1) factors contributing to MLC preference; (2) Thai public schools as a preferred choice for long term residents; (3) implicit discrimination; and (4) unresponsive policy and practice. The code structure is presented in Table 3. Key descriptions of each theme are as follows.

### 3.1. Factors Contributing to MLC Preference

Many factors contributed to parental education choices, including cultural and language preference, school location, household income level, and parental education background and perspectives. MLCs appeared to be a preferable choice over TPSs in most migrant parents. Although parents need to pay tuition fees in MLCs, most parents still chose MLCs (2300 children) over TPSs (900 children). The local MLC officer collected these figures. All 13 MLCs were established in 1999 by local NGOs. In the past, Thai communities blamed migrant children for local petty crime and burglary. To address these problems, NGOs discussed with local villagers and agreed to set up 13 MLCs to serve as community nurseries or educational spaces to change, as the villagers perceived it, the children’s misconduct. MLCs gradually evolved and are now operating as ‘schools’, providing grade 1 to grade 12 education depending on the MLC policies. 

Cultural preference plays an important role in parental decisions. Most interviewees (M3, M7, M9, and M15–M19) revealed that MLCs had some distinctive features that seemed to be attractive for migrant families, such as Myanmar curricula and teaching by native Myanmar instructors. Most MLCs are located close to migrant communities and serve as ‘close-to-clients’ nurseries and schools, rendering a lower transportation cost on migrants. Some MLCs made an internal agreement with Myanmar schools so that graduates from MLCs can continue their study in Myanmar.
“*Some parents are incapable of sending their children to standard schools. There is a transportation fee, so it would be better to study at the nearby MLCs, because the main expectation is just to be literate. Sending them to MLCs is safer; at least it provides a day-time shelter, while parents are working.*”—M04

Myanmar parents, who aim to return to their homeland in the future, or wish to have their children work in Myanmar after being brought up, tended to choose MLCs for their children. This finding is consistent among all interviewees. Migrant children in Mueang district were mostly living in migrant-populated communities, while migrants in Kraburi district tended to live close to Thai communities; this is due to the fact that most migrants in Kraburi district were employees in rubber fields owned by a Thai employer. This meant most migrant children in Mueang district were not fluent in the Thai language. Sending children to MLCs also helped cut transportation cost as most MLCs are situated close to communities (despite extra costs for tuition fees). 

Another contributing factor to the preference for MLCs is the ‘openness’ and ‘flexibility’ of MLCs in ‘document checking’ compared to TPSs. Although the ‘citizenship’ or ‘legal’ status of a child’s parents is not a prerequisite to enter TPSs, some interviewees such as E06 and M19 mentioned that TPS’ teachers usually asked for official documents, such as work permits or passports. This made some parents who had precarious legal status reluctant to send their children to TPSs. Parents who tried to register sometimes had to pay a large amount of money which finally affected the family’s financial status. In contrast, MLCs were quite open to ‘anybody’. Document checking was less strict, and MLCs were perceived by parents as ‘more open’ to migrants.

Therefore, the parents of most migrant children in Mueang district were more likely to send their child to MLCs instead of TPSs.
“*I chose MLCs, because I wanted my children to continue their higher education and get a job in Myanmar in the future. If there is no MLC here, I would rather send them back to Myanmar.*”—M01


Not all parents are supportive. Some parents, especially those with lower educational backgrounds, did not encourage their children to have higher education. They thought that children aged around 12–13 years (regardless of sex) were already capable for work, and were therefore meant to get a job and earn money to support the family. However, some parents (M04, M06, M07), particularly the educated ones, expressed that having children educated was worth the money.
“*If there were no MLCs, half the parents might choose to send their children to Thai schools; however, the other half would rather let their children stay at home as it is unaffordable. If parents are educated, they do everything to educate their children. On the other hand, less educated parents think that their children can work, so it is better to work and earn more money.*”—M04

However, running MLCs is not always easy. One of the critical challenges is that MLCs are not based on a legal ground like most TPSs. There were no official ‘school standards’ imposed on MLCs from the outset, or funding support from the Government. MLCs staff (E07, E08) thought that MLCs faced many operational problems, such as a poorly ventilated environment, lack of school lunch or milk provision, and lack of certified teachers. All MLCs were run solely on tuition fees collected from migrant parents (around 100–200 Baht each, varying from one MLC to another), plus cash support from NGOs founders.
“*I’ve spoken to the teachers that we should limit the number of student, as this is too crowded. Then I would say that this place is more like a nursery, than school. They’ve got limited budget from funders, so we need to admit that the tuition fee is also their main financial source. This is why we can’t limit the number of students; it is a vicious cycle and becoming more business oriented.*”—E04

### 3.2. Thai Public Schools: A Preferred Choice for Long-Term Stayers

As mentioned in the introduction, TPSs are free for all children in Thailand, regardless of their nationality. This policy was devised in the Cabinet Resolution, ‘Rule of the Ministry of Education on Evidentiary Document for Pupils and Students Admission into Establishment of Education B.E. 2548 (2005)’ on 5 July, 2005 [10]. The policy also opened up education to migrants who did not have any proof of identity. In detail, the policy allows TPSs to generate an ‘identity code’ to migrant students who do not have a national identity number like Thai national children. This code starts with ‘G’, and undocumented migrant children attending TPSs will acquire a unique identity code stating with G (such as G1257627083). This number is recognized by the education sector only, not by the civil registration authorities [16]. The school directors normally send the figure of ‘G-series’ children to the Ministry of Education (MOE) at the beginning of the academic year to ask for budget support. The MOE will finance TPSs on a capitation basis (around 1900 Baht per individual).

Migrants send their children to TPSs for a range of reasons. Parents who planned for a long-term stay in Thailand tended to send their children to TPSs. Some parents wished their children to be more fluent in Thai, and therefore TPSs were a desirable choice in this respect.
“*I will stay in Thailand for quite a long period, probably not going back to Myanmar. Then I want my children to study in a Thai school, so they may continue to higher education here.*”—M03
“*We work in Thailand, but we cannot read or speak Thai. We are at a disadvantage. We wanted our daughter to be literate in Thai, so she may help us when we need to communicate in the Thai language*”—M06 (interviewee and husband)

Some families were living in Thailand for a long time but failed to register as ‘Thai nationals’. This population is de facto not migrant. Ranong residents usually called them displaced Thais. Many of them still did not have a civil identification number. This situation made their status quite similar to undocumented migrants (though in reality they were not). The absence of a civil identification number hampered access to education amongst displaced-Thai children. One interviewee admitted that he had asked his Thai friend to act as legal guardian of a displaced-Thai child in order to enable that child to register as a Thai national. Displaced-Thai children are allowed to enter TPSs for free (like Thai nationals and migrant children). However, problems still remain as the free education service is only for basic education while some displaced-Thai parents wish to have their child continue to high-school or to bachelor-degree education.
“*I begged my Thai friend for his surname (to act as legal guardian), I need it for my children. Luckily, he felt pity for them and for the sake of their future, he let them use it.*”—M19

### 3.3. Implicit Discrimination

The Kingdom of Thailand offers free mandatory education to every child; however, some implicit marginalization remains. Interviewees who worked as school-teachers in TPSs mentioned that the number of Thai students was continuously declining. This was consistent with the document review by the Office of the National Economic and Social Development Board which suggested that the infertility rate in Thailand was constantly decreasing, from almost 1 million new-borns per year in 1993 to 680,000 in 2017 [17]. Many schools in migrant-populated areas tend to accept more migrant students, because they want to avoid having fewer than 40 students; this would require them to merge with bigger schools [18]. 

Some interviewees found that Thai parents viewed schools with migrant students to be of lower quality; and consequently, some Thai parents took their children to other schools that had only ‘Thai’ students.
“*At the beginning, Thai students were the majority in the school. Later when we updated the policy to include Myanmar students, because we did not want our school to merge with another, there was resistance from Thai parents. They did not want their kids to assimilate with Myanmar students.*”—E06

TPSs teachers expressed their view towards Myanmar students in another direction. The interviewees (E05, E06) mentioned that Myanmar children usually had poor hygiene at the beginning of the semester. However, they patiently taught the students to be disciplined in person and in school (via many activities, such as classroom cleaning duty), and the students’ hygiene behaviour improved tremendously. The teachers even admitted that Myanmar students were more hardworking than Thai students.
“*We did not mind that they are Myanmar students, they are all our students. They did not cause any trouble and now that they are our students, whatever their actions are, they reflect our qualities.*”—E05

The social environment in schools created some difficulties for Myanmar students. The interviewees (M10 and M11) mentioned that they noticed some migrant children facing verbal or even physical bullying from their school peers of a different race, religion and language spoken. Myanmar students usually were blamed if there was a fight with Thai students. Sometimes they suffered name-calling by ‘Myanmar’ instead of their own name, and teachers also encountered bullying words towards migrants. Some bullied students eventually resigned from the schools.
“*I know that she can continue education, but she came to me and said that she wanted to quit, as she was called as ‘Myanmar’ all the time.*”—E16
“*Normally my son is safe in front of the teacher, but he was thrown a rock when teachers were away. The previous day, we were in a queue for snacks, suddenly one Thai Dad commanded us to get out of the queue, and brought his son in line instead.*”—M11
“*I must admit that there is social pressure, sometimes people called me ‘Myanmar Students Director’.*”—E05

### 3.4. Unresponsive Policy and Practice

The 1999 National Education Act, Section 10 of Chapter 2 on Educational Rights and Duties indicates that, “In the provision of education, all individuals shall have equal rights and opportunities to receive basic education by the State for the duration of at least 12 years. Such education, provided on a nationwide basis, shall be of quality and free of charge” [19]. The Ministry of Education provides financial support to every student on a per-head basis. The support includes (1) tuition fee; (2) book fee; (3) Stationery fee; (4) Student uniform fee; and (5) Extra curriculum activities fee. Despite the existence of such a policy, most interviewees mentioned that the policy per se might not be successfully implemented due to certain reasons. First, there is the cultural difference as described in the earlier section. Some migrant families still preferred to send their children to MLCs rather than TPSs as the Myanmar language was not normally taught in TPSs. This point was also coupled with the fact that the Thai education system is not linked with the Myanmar education system. Graduating from (official) TPSs does not guarantee eligibility to continue to study in Myanmar. 

Secondly, some interviewees such as M08 and E06 also raised concerns about uneducated children, who attended neither the TPSs nor the MLCs. A concrete statistical number of uneducated children is missing. However, the number of uneducated children might be almost half of the total number of migrant children, as expressed by interviewees M08 and E06. Key reasons for leaving children uneducated were (1) attitudes of parents that wish to have their child get a job and earn as early as possible rather than attending school; and (2) the fear that having their children attend schools might incur an expense. 

“*Approximately 30 percent may be uneducated, however in this community the number may be as high as 50 percent.*”—M08

Health promotion policies at schools also seemed problematic. Normally, local health practitioners at the health centre are responsible for school health activities in his or her ‘catchment’ areas but activities were confined to TPSs and not MLCs. This is because the budget allocated at the beginning of every fiscal year was calculated based on the ‘official’ TPSs, while MLCs were not counted from the outset. In practice, both Thai and non-Thai children usually visited health centres when they were sick. Local nurses (H05–H07) complained of a huge work burden, which did not match with the budget provided. Dental care was one service obviously overlooked. However, basic vaccinations are provided free of charge to all migrant children, despite the absence of written policy indicating this. In practice, the amount of vaccines supplied to health centers is usually calculated from the number of ‘Thai’ children in the catchment area plus 10% surplus as a margin. The utilization rate of vaccinations in children was quite low, therefore, the local providers often used that surplus to vaccinate migrant children.
“*MLCs are all Myanmar children; they are out of our scope and it is out of our hands to look after them. We may conduct some health promotion campaigns, but we really cannot offer adequate fundamental school health services for them all.*”—H05
“*Dental check-ups here is for four to five Thai students; the Myanmar students just watch, they understand their limited rights.*”—E05
“*Vaccination in migrant children is exactly the same as in Thai children, we barely distinguish race and ethnicity in this issue. As long as we have the stock and all children received them, then that is good enough.*”—H07

## 4. Discussion

Access to education services among migrant children is influenced by a number of factors, and this issue has now become an international concern. A survey of migration policies in 28 countries by Klugman and Pereira showed that only 40% of developed countries and over 50% of developing countries did not allow children with irregular status access to schooling [20]. The UNICEF report in 2017 also demonstrates that in several countries such as Germany, entry into the school system for migrant children is often determined by many administrative barriers such as the rules of the federal state where they land and their prospects for permanent residence in Germany [21]. In South Africa, there is inconsistency between the legislative frameworks on school admission for migrant students and what the schools implement in reality. Crush and Tawodzera reported that some public schools usually demand study permits and birth certificates which are often difficult to obtain in migrant children with precarious residence status [22].

Such situations also occur in Thailand, as evidenced by the results above. The existence of education policy does not mean that migrant children will enjoy basic education rights, including health promotion activities at schools. 

The following topics are cross-cutting themes and constructs that help explain migrant education in Ranong, that is: (1) the difference between de jure policy design and de facto implementation, as a consequence of cultural and language differences; and (2) the Migrant Learning Centre, a vacuum situation.

### 4.1. Difference between De Jure Policy Design and De Facto Implementation—Consequence of Cultural and Language Differences

De jure, all children on Thai soil are supposed to have equitable access to education. De facto, some migrant children are left uneducated or experience a sub-standard education system. This phenomenon is due to several factors. A low socio-economic status of migrants and language difference has played an important role. For instance, migrant children despite being eligible to study in TPSs, need to have basic Thai literacy first. This requirement is not written, but is widely accepted and followed. Another socially excluded criteria is household income, as although the Government subsidizes tuition and other fees, parents still need to pay for transportation costs, which is sometimes unaffordable [23]. Additionally, the routine practice of TPSs that thoroughly check the proof-of-identity documents of migrants’ parents undermined their willingness to send their child to TPSs. These accounts are exemplary explanations why TPSs are not the preferred choice of education as intended by the policy.

In addition, parental attitudes did have some influences. Some parents wished to get their children in the labour force as early as possible rather than send them to school. The finding that girls are susceptible to opt out of education due to marriage, is not obvious in this study, although other studies listed this as a concern [24,25].

Displaced Thai participants had different perspectives of education. They considered themselves as Thai citizens who wanted to reside in Thailand permanently, and enthusiastically wanted their children to continue in higher education in Thailand. Although they did not report discrimination in their communities, it remains a legislative matter. A displaced Thai father needed to beg for his friend’s surname, in order to allow his children to continue at higher education. Although the Government has expanded education and health coverage for displaced Thai people, the lack of policy consistency and continuity remains [26].

Ethnic and language disparities also made migrant children vulnerable to bullying, which contributed to school attrition. Unexpected social discrimination as a result of poor attitudes of Thai parents was also experienced. Thai schools that follow Government policy by accepting migrant children are viewed as low-quality schools, but in spite of this, many schools are still willing to enroll migrant children. This finding contrasts with a prior study from Punpuing et al. (2014), which reported that teachers were reluctant to accept migrant children. This was because school teachers feared that school performance (as a whole) might get worse due to the poor academic performances of migrants and that the drop-out rate might increase due to the highly mobile behaviour of migrants [27]. The likely explanation for the different findings is that most TPSs in Ranong have only a few Thai pupils; therefore, enrolling migrant children is an effective strategy to obtain additional budget from the Government. 

Many examples in this study are evidence that cultural, language and citizenship differences between migrants and host populations influentially determine the degree of access to public services. This point is supported by international literature, such as Hultsjo (2005), Rosenberg (2006), Manirunkunda (2006) and Worth (2009) [28,29,30,31].

The difference between de jure policy intention and de facto policy implementation is seen in education and also health, particularly health promotion and disease prevention. The Ministry of Public Health has set its vision that all people on Thai soil should be able to enjoy health care access without catastrophic payment incurred [32], yet there are some observed adaptive behaviours among healthcare providers, which can be positive or negative. The positive is exemplified in the case of vaccines where providers provided vaccinations to migrants despite the absence of written documents [33]. The negative is seen in dental care where migrant children were the last priority. This was explained by the fact that the health-promotion budget received is calculated from the volume of Thai nationals, not migrants. The case was more severe in MLCs which are out of reach of the health providers at sub-district health centers. This is because MLCSs are not perceived as ‘schools’ in the view of public providers. This phenomenon is linked to the concept of street-level bureaucracy as proposed by Lipsky, which indicates that frontline staff sometimes adapt their behaviour towards clients due to the insufficiency of resources and pressure from society [34]. One of the distinct adaptive strategies is ‘prioritizing’ clients, as reflected in the case of dental care in TPSs and all health promotion services in MLCs.

### 4.2. Migrant Learning Centre—A Vacuum Situation

The number of migrant children in MLCs is more than twice those in TPSs. Our findings underpin the importance of cultural and language preferences on school choice. Similar circumstances occurred in Tak, another migrant populated province in Thailand, which has around 65 MLCs with more than 8000 migrant students in total. The number of MLCs in Tak has increased from 3 to 65 in 16 years [35]. This evidence emphasizes the preference of MLCs amongst migrant populations in Thailand. 

However, it appeared that the MLCs in Ranong province were sub-standard with a lack of nutritional support, inadequate school lunch and milk provision, and poorly ventilated buildings which all contributed to unhealthy conditions for children. Non-certified teachers, a lack of supervision and limited budget are also other important factors that cause MLCs to be locked in a ‘vacuum’ of migrant education policies in Thailand. This point is consistent with a recent report by Save the Children which highlights that the quality of education across MLCs is highly inconsistent, and most MLCs are suffering a lack of centralized oversight, financial instability and limited resources [11].

More importantly, considering the sub-standard teaching, it might be worth questioning if MLCs ‘teachers’ can be called ‘teachers’ in the view of Thai officials. Another question arises as to whether so called ‘teachers’ could be prosecuted for engaging in a job that is not specified in their work permit [20]. According to a Cabinet resolution in 2014, Cambodia–Laos–Myanmar (CLM) migrants registered with the One Stop Service are allowed to work lawfully in two occupations: construction labour and housemaids [36]. 

Recently, the Provincial Education Office announced rules and regulations for MLCs. These regulations are like an internal agreement within Ranong only. The objective of imposing rules is to raise education standards and to supervise the social welfare of children so that no one can take advantage of children through the face of ‘education’. However, some rules seem impractical for MLCs such as: the use of Thai curricula; that all teachers must be officially certified (completing bachelor degrees on pedagogy); and that all MLCs need to pass a school registration process [23,35]. The most interesting (and also intriguing) question is who will be the law enforcer of these rules. This is because MLCs were not officially established within the legal instrument of any ministry such as the Ministry of Education, Ministry of Public Health, or Ministry of Social Development and Human Security. As a result, no authority takes full responsibility over MLCs or the more than 2000 migrant children. It is no exaggeration to say that MLCs are in the vacuum of education policies for migrants in Thailand, as illustrated in Figure 1.

This study by no means intends to indicate that the education policy for migrants is a ‘failure’. In fact, Thailand has made huge progress in ‘ratifying’ the rights of education for everybody on Thai soil, by the instigation of the Cabinet Resolution in 2005. However, the points raised in this study show that there are still challenges that need overcoming when the policy is implemented in reality; in this sense, there is room for improvement. MLCs seem to be more responsive than TPSs to the needs of migrants’ families. However, MLCs are left unsupervised and are encountering a lack of resources, in both funding and qualified workforce. The Thai Government may need to consider MLCs as ‘allies’ in expanding education (and also school health) coverage to ‘all children’. Nevertheless, MLC supervision should be tailor-made, taking into account the State’s perspectives (such as national security and standardized quality across schools) and the needs of users (such as cultural and language preference). Imposing a ‘one-size-fits-all’ regulation on MLCs, similar to TPS, might not be an effective and sustainable solution.

To elaborate more on this point, the Thai government needs to collaborate more with the MLCs. Education standards should be ensured, along with fundamental school hygiene. However, regulation should be done with respect of the essence of cultural perspectives, language differences, and core values of the MLCs. These efforts should be implemented in a collaborative fashion rather than in an authoritative fashion. A participatory public policy process that engages all stakeholders— including officials from MOE, health care providers, NGOs, and migrants themselves—to take part in policy dialogue is needed in order to find ‘acceptable’ solutions for all. With such dialogue, although the ‘final solutions’ cannot be achieved, at least everybody will know what the gaps in knowledge are and what should be done to fill those gaps. A number of potential research questions arise from this study, for instance: (1) What is the actual number of ‘uneducated’ migrant children in Thailand? (2) What is the social impact of leaving them uneducated? (3) How can Thailand and Myanmar collaborate with each other to create a seamless education system with acceptable standards for both countries? (4) What are the gaps in current employment laws that might affect the hiring of foreign teachers in Thailand? and (5) Is the provision of dental services to all migrant children cost-effective? If so, is it worth the budget invested?

Despite a rigorous study design and richness of information, this study still has some limitations; examples are described as follows. The first is about the world views of the researchers themselves. The research team had extensive experience in migrant health studies and research on vulnerable populations; therefore, the familiarity with migrants might make the interpretation of the results lean towards a pro-migrant side. However, the researchers avoided this bias by triangulating the interview data by various means, such as cross-checking interviews across respondents or using document review to support or objectify the interview data. The second is acquiescence bias due to the expectations from both interviewees and researchers, which might distort the given information. The interviewees knew the status of the researchers to be civil servants from the MOPH. Therefore, they might provide biased information. Lastly, as this is a case study in a single province, generalizing the results to other settings should be done with caution. There are a number of non-Thai populations in Thailand and this study focused only on ‘CLM migrants’ (with some small discussions on displaced Thais), while other non-Thais, such as tourists, expats, and refugees, have been unexplored.

## 5. Conclusions

Access to education for migrant children in Ranong is influenced by numerous factors. MLCs are the favourable choice of education over official TPSs in most migrant families. Cultural and language differences played pivotal roles in selection choices of education. The MLCs are usually confronted with low resources and a lack of supervision of educational standards and hygienic environment. Health promotion activities are not routinely performed in MLCs. Dental service is under-serviced in most MLCs and TPSs. Implicit and explicit discrimination towards migrant children was observed.

The Thai Government should view MLCs as allies in expanding education coverage to all children in the Thai territory. A participatory public policy process that engages all stakeholders, including education officials, health care providers, NGOs, MLCs representatives, and migrants themselves is needed to improve the education standards of MLCs and, at the same time, still keep their culturally-sensitive strengths.

## Figures and Tables

**Figure 1 ijerph-16-00430-f001:**
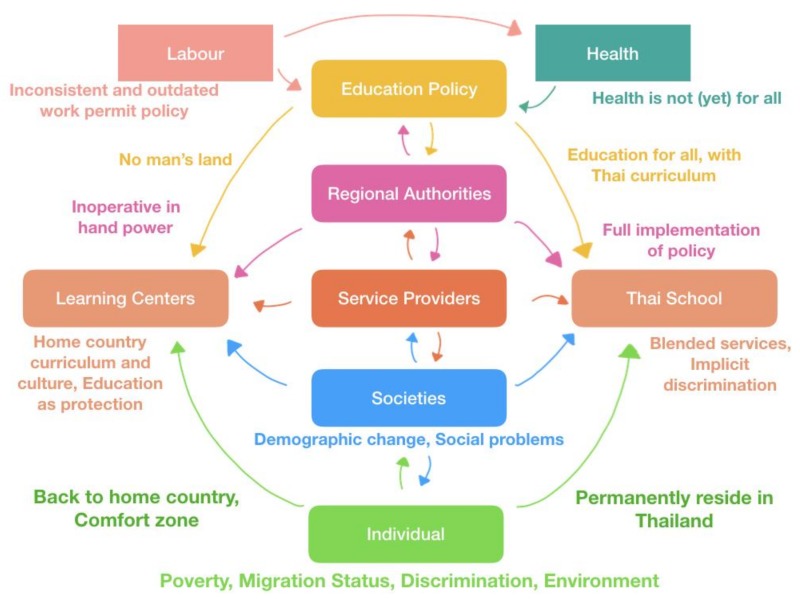
Relationships and determinants of health and access to education in migrant children.

**Table 1 ijerph-16-00430-t001:** Participants’ characteristics.

Code	Age	Race	Gender	Class	Code	Age	Race	Gender	Class
E01	50–60	Thai	F	Central policy maker in Ministry of Education	M03	40–50	Myanmar	F	Migrant parent in city center
E02	50–60	Thai	M	Provincial policy maker in Ministry of Education	M04	50–60	Myanmar	F	Migrant parent in city center
E03	50–60	Thai	M	Headquarter NGO manager	M05	40–50	Myanmar	F	Migrant parent in city center
E04	30–40	Thai	F	Provincial NGO staff	M06	40–50	Myanmar	F	Migrant parent in city center
E05	50–60	Thai	F	Director of Thai state school 1	M07	30–40	Myanmar	F	Migrant parent in city center
E06	40–50	Thai	F	Teacher of Thai state school 2	M08	20–30	Myanmar	F	Migrant parent in city center
E07	40-50	Thai	M	Secretary of Migrant Learning Center 1	M09	30–40	Myanmar	F	Migrant parent in city center
E08	30–40	Myanmar	F	Secretary of Migrant Learning Center 2	M10	40–50	Myanmar	F	Migrant parent in city center
H01	40–50	Thai	F	Central policy maker in Ministry of Public Health	M11	30–40	Myanmar	M	Migrant parent in city center
H02	40–50	Thai	F	Central policy maker in Ministry of Public Health	M12	30–40	Stateless	F	Stateless parent in Kraburi
H03	40–50	Thai	M	Staff of Ranong Provincial Public Health Office	M13	30–40	Stateless	F	Stateless parent in Kraburi
H04	40–50	Thai	F	Staff of Ranong Provincial Public Health Office	M14	40–50	Stateless	M	Stateless parent in Kraburi
H05	40–50	Thai	F	Nurse of Health Promoting Hospital 1	M15	40–50	Stateless	M	Stateless parent in Kraburi
H06	40–50	Thai	F	Nurses of Health Promoting Hospital 2	M16	40–50	Stateless	F	Stateless parent in Kraburi
H07	30–40	Thai	F	Nurses of Health Promoting Hospital 2	M17	40–50	Stateless	F	Stateless parent in Kraburi
M01	30–40	Myanmar	F	Migrant parent in city center	M18	30–40	Stateless	M	Stateless parent in Kraburi
M02	40–50	Myanmar	F	Migrant parent in city center	M19	40–50	Stateless	M	Stateless parent in Kraburi

Abbreviations: F—Female; M—Male; NGO—Non-governmental organization.

**Table 2 ijerph-16-00430-t002:** Question guide for interviewers.

Question Guide for Policy Makers	Question Guide for Local Officers	Question Guide for Migrants
Please tell me about your position [How long have you been in this position? What about your past experience in this work? What are the role and responsibility of your job regarding migrant healthcare and education policies?].	Please tell me about your job [How long have you been in this job? What about your past experience in this job?].	Please tell me about yourself [Please describe more about your occupation, how long have you been here in Thailand?].
Please briefly explain how you have been involved in the development of health insurance policy and migrant education policy [How was it developed? Who has been involved? What was the original intention/goal of the policies?].	Please tell me about your daily job with regards to migrants [Do you have many migrants coming to your facility each day? How many migrant children in your school?].	Please tell me about your family [How many family members are there in your family? What are their occupations?].
Now that the insurance policy and education policy for migrants have been implemented already, what are your opinions towards the policies?	What are problems that you experience in dealing with migrant patients/students? [What about the legal status problem? Is there any problem about the language barrier? What about the cost of treatment/education of illegal uninsured migrants?	How did you come to be working here in Ranong? [Please describe more about how you came into the country. Who helped you settle down in Thailand?].
Please tell me about the positive sides and the negative sides of the policies [What are the key challenges? How can those challenges be overcome?], and please suggest ways for further improvement.	In your opinion, before and after the insurance/education policy, are there any changes to migrants’ access to care/education? Please tell me more about your perceptions on this issue?	How do you support your family? [Please tell me about the estimated monthly income of your family and the estimated monthly expenses].
Please tell me your perspectives about the health/education accessibility of the migrant children? Were the policies well prepared or did the implementation work well in reality? Are the services provided adequate and equitable?	Please tell me how you know about the policy [From which routes/channels (official document from the ministry, attending workshop, being informed by peers?)].	According to the income levels, have you ever experienced discrimination or inaccessibility to healthcare or education for your children?
Please tell me what would you see as barriers to health/education accessibility of the migrant children? Were they well managed with the current policies? If not, will there be future efforts to manage the barriers?	Has the insurance/education policy made any impact on your daily work [No change? or significant change? What about any additional burdens?].	Please tell me about your experience in taking your children to the health facility/school, please identify type of healthcare facility; public or private.
In your perspectives, do migrant children experience discrimination or privilege in health or education?	Have you ever experienced any constraints in your work with regards to the policy? Please explain more about that situation and how you cope with it.	What are your experiences as a female/male migrant in accessing healthcare? Are there any specific challenges you face?
	What do you think about the policy guideline from the ministry [Does it work? If so, or if not, why do you think accordingly?].	How has your identity as a female/male migrant (or your child’s identity) affected your ability to access healthcare/education?
	Please tell me how the MOPH/MOE communicates with your institution [Any documents sent to and from the ministry regularly? Any workshops or consultative meetings held by the ministry? How did you give feedback about your concerns to the ministry?	Have you ever felt discriminated against because of being a poor migrant? Explain OR Have you ever been poorly treated due to your identity as a female/male migrant?
	Who else do you have to work with in running the policy? [Ministry of Labour, Ministry of Interior, NGOs] What is your experience in working with them?	Have you experienced any privileges when accessing healthcare services/education because of being a poor migrant?
	To what extent does the policy design fit your local context?	Do you feel that there are people who are given certain privileges when accessing care and some are not? Explain.
	In your view, what are the benefits and downsides of this policy?	Do you feel schools/health facilities fulfil your healthcare needs as a poor migrant? Explain.
	Please tell me your suggestions how the policy should be improved in order to better fit your local context.	Was there anything done at public facility/school that made you uncomfortable in receiving services? Please describe.

**Table 3 ijerph-16-00430-t003:** Thematic analysis of the study.

Themes	Categories	Codes	Interviewees
(1) Migrant Learning Centres (MLCs) preference	MLC	(1) MLC establishment and its characteristics; education as Protection	E02–E07, M01, M02, H03, H04, H06, H07
		(2) Resources for MLC and unreachable standards	E02–E04, E06, E07, H03, H04
	Curriculum incoherence	(3) Incorporate curriculum with discontinuity of education	E02–E07, H05, M3, M7, M9, M15–M19
	Migrants perspectives	(4) Parents’ perspectives on children’s needs	E01–E07, M1–M19, L01
		(5) Displaced-Thai: citizenship struggle	M15, M18, M19
		(6) Comfort zone within migrant communities and uneducated children	E02, E03, E06, M01–M07
	School Hygiene Standard	(7) Health-promoting school standards	H01, H02, H05
		(8) Aids upon request	E02, H01–H07
	Unprepared Resources	(9) Limited budget and immobilized existing resources	H03–H07
		(10) Limited human resources	H03–H07
(2) Thai Public Schools for long term residents	School Hygiene Standard	(7) Health-promoting school standards	H01, H02, H05
	Demographic Change	(11) Low birth rate among Thai and demographic change	E01–E03, E05, E06, M05, H03–H05, L01
	Operational Level	(12) No obvious discrimination within Thai Schools	E01, E02, E04–E06, M15, M18, M19
		(13) Unified school health services	E06, H03–H07
	Remnant of Discrimination	(14) Society and school discrimination with unwritten admission criteria	E01–E06, M03–M11
		(15) Acceptance of policy discrimination	M19
(3) Implicit discrimination	School Hygiene Standard	(7) Health-promoting school standards	H01, H02, H05
	Remnant of Discrimination	(14) Society and school discrimination with unwritten admission criteria	E01–E06, M03–M11
		(15) Acceptance of policy discrimination	M19
	Operative Power	(16) No man’s land with inoperative in-hand power	E01–E04, E07, M05
	Access to healthcare	(17) Inadequate access to healthcare services; especially dental care	E03–E05, E07, H05–H07
(4) Unresponsive policy and practice	MLC	(2) Resources for LC and unreachable standard	E02–E04, E06, E07, H03, H04
	Curriculum incoherence	(3) Incorporate curriculum with discontinuity of education	E02–E07, H05, M3, M7, M9, M15–M19
	Migrants perspectives	(5) Displaced-Thai: citizenship struggle	M15, M18, M19
	Operative Power	(16) No man’s land with inoperative in-hand power	E01–E04, E07, M05
	Policies Discordance	(18) Fuzzy policy and barrier to policy communication	E01–E03, E05–E07, M03, M15–M19, H03, H05, H06, H07, L01
		(19) Labour policy effects on health and education	E01, E02, E05, M03, M18, M19, H03
	Access to healthcare	(20) School hygiene efforts	E01–E05, E07, M01–M03, M15, M17, M19
		(21) Healthcare insurance in reality	H03, H04, H06, H07

Remarks: Codes were developed into categories and themes as: Theme 1: codes 1–10; Theme 2: codes 7, 11–15; Theme 3: codes 7, 14–17; Theme 4: codes 2, 3, 5, 16, 18–21. All details were listed in results section.

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
