# Peer review of "Education Policy for Migrant Children in Thailand and How It Really Happens; A Case Study of Ranong Province, Thailand"

_ijerph, 2019, doi:10.3390/ijerph16030430_

Reviewer 1 Report

This is a well-written, well-researched paper that appears to fill a gap in the scholarly literature, namely, about why Myanmar parents choose to send their children to a Migrant Learning Centers (MLCs) or a Thai Public Schools (TPSs). The research design and execution all appear to meet appropriate scholarly standards. The topic will interest many education policy researchers, governments, and NGOs.

I would encourage the authors to add a few more paragraphs talking about their recommendations. "The Thai Government may need to consider MLCs as ‘allies’ in expanding education (and also school health) coverage to ‘all children’. Nevertheless, MLCs supervision should be tailor-made, taking into account the State’s perspectives (such as national security and standardized quality across schools) and the needs of users (such as cultural and language preference). Imposing a ‘one-size-fits-all’ regulation on MLCs, similarly to TPS, might not be an effective and sustainable solution."

On the one hand, the MLCs are attractive beyond they are not controlled by the state. On the other hand, the state may want to regulate MLCS to improve, or at least standardize, quality. But can you have it both ways? If Myanmar parents want their children to learn the language, won't regulation just mean that MLCs will need to teach Thai? Won't regulating MLC's effectively just make them TPSs, for good and ill?

A few questions, then:

What do Myanmar parents and educators think about the plan to regulate MLCs? Would they go along with this? Or would they oppose it?   

What does the Thai government want to do about MLCs? Do they want to regulate them? What do they have in mind, eg, about language and curriculum? 

Are international actors relevant in this story? Do Western NGOs or the UN want to intervene in this debate?

I would also, finally, encourage the authors to situate this story in a larger academic debate about the global education reform movement.

Author Response

Response to Reviewer 1

I would encourage the authors to add a few more paragraphs talking about their recommendations. "The Thai Government may need to consider MLCs as ‘allies’ in expanding education (and also school health) coverage to ‘all children’. Nevertheless, MLCs supervision should be tailor-made, taking into account the State’s perspectives (such as national security and standardized quality across schools) and the needs of users (such as cultural and language preference). Imposing a ‘one-size-fits-all’ regulation on MLCs, similarly to TPS, might not be an effective and sustainable solution."

On the one hand, the MLCs are attractive beyond they are not controlled by the state. On the other hand, the state may want to regulate MLCS to improve, or at least standardize, quality. But can you have it both ways? If Myanmar parents want their children to learn the language, won't regulation just mean that MLCs will need to teach Thai? Won't regulating MLC's effectively just make them TPSs, for good and ill?

-Thank you for the recommendation. We have added the discussion part in Line 354- 369 and 497- 501. However, the recommendations were made based on Thai cultural aspects and political terrain. Further policy recommendations maybe evaluated in the future studies, as the data from this article can derive the policy recommendations as written.

A few questions, then:

What do Myanmar parents and educators think about the plan to regulate MLCs? Would they go along with this? Or would they oppose it?   

- Myanmar parents accept with the current MLCs standard; however, the educators’ views are beyond our study. According to the Thai local officers, Myanmar educators try to standardized MLCs by involving them into Myanmar distance learning scheme.

What does the Thai government want to do about MLCs? Do they want to regulate them? What do they have in mind, eg, about language and curriculum? 

- Thai government wants to standardized them, however, limited budget from both government and MLCs were found. Which lead to the ‘overstandardized’ issue mentioned in the early of this article. Therefore, MLCs is now out of scope for Thai government.

Are international actors relevant in this story? Do Western NGOs or the UN want to intervene in this debate? 

- According to my current knowledge, the international NGOs or UN are not so involved in this area. There are some involvements of international NGOs in Mae-sot district, but not in Ranong province. Which lead to inequity for migrant children in both sides. The UN is not involved in the MLCs issue for this point of time.

I would also, finally, encourage the authors to situate this story in a larger academic debate about the global education reform movement.

-Thank you very much, the authors also hope to expand the issue of MLCs education and school health to a wider arena.

Reviewer 2 Report

The article offers a timely discussion of the challenges of educational policy as seen in the Thailand context in relationship to migrant children. Some of the inner dynamics concerning local attitudes toward migrant families, the difference between short-term and long-term residency and the difference between Thai public schools and the privately run migrant schools is clearly presented. Having worked with NGO-managed schools in South Asia, the data and analysis presented was enlightening. It is hoped that local officials can see this data and the conclusions so that they may improve the situation for migrant children. The concerns about dental care was especially noted. The English was very good. I have attached a document that notes simple edit suggestions.

Author Response

Response to Reviewer 2

Thank you very much for your response. We have compiled all the comments as in the attached file.

Titiporn T

On behalf of the authors team
